# Voltammetric analysis for distinguishing portal hypertension-related from malignancy-related ascites: A proof of concept study

**Moises Muley** [1], **Umberto Vespasiani-Gentilucci**[1], **Antonio De Vincentis**[1]*, **Marco Santonico**[2], **Giorgio Pennazza**[3], **Simona Sanguedolce**[1], **Cristiana De Luca**[4], **Francesco Plotti**[4], **Antonio Picardi**[1], **Raffaele Antonelli-Incalzi**[5]

**1** Internal Medicine and Hepatology, Università Campus Bio-Medico, Roma, Lazio, Italy, **2** Unit of Electronics for Sensor Systems, Department of Science and Technology for Humans and the Environment, Università Campus Bio-Medico, Roma, Lazio, Italy, **3** Electronics for Sensor Systems Unit, Department of Engineering, Università Campus Bio-Medico, Roma, Lazio, Italy, **4** Gynaecology and Obstetrics Unit, Università Campus Bio-Medico, Roma, Lazio, Italy, **5** Geriatric Unit, Università Campus Bio-Medico, Roma, Lazio, Italy

* a.devincentis@unicampus.it

## Abstract

### Background

Serum-ascites albumin gradient (SAAG) remains the most sensitive and specific marker for the differentiation of ascites due to portal hypertension from ascites due to other causes. SAAG has some limitations and may fail in selected conditions. Voltammetric analysis (VA) has been used for the detection of electroactive species of biological significance and has proven effective for detection infections in biological fluids.

### Aims

In this study, we compared the accuracy of voltammetric analysis (VA) with that of SAAG to differentiate ascites due to portal hypertension from that having a different origin.

### Methods

80 ascites samples were obtained from patients undergoing paracentesis at the Campus Bio-Medico Hospital of Rome. VA was performed using the BIONOTE device. The ability of VA to discriminate ascitic fluid etiology and biochemical parameters was evaluated using Partial Least Square Discriminant Analysis (PLS-DA), with ten-fold cross-validations.

### Results

Mean age was 68.6 years (SD 12.5), 58% were male. Ascites was secondary to only portal hypertension in 72.5% of cases (58 subjects) and it was secondary to a baseline neoplastic disease in 27.5% of cases (22 subjects). Compared to SAAG$\geq$1.1, e-tongue predicted ascites from portal hypertension with a better accuracy (92.5% *Vs* 87.5%); sensitivity (98.3% *Vs* 94.8%); specificity (77.3% *Vs* 68.2%); predictive values (PPV 91.9% *Vs* 88.7% and NPV 94.4% *Vs* 83.3%). VA correctly classified ascites etiology in 57/58 (98.2%) of cases with

**Data Availability Statement:** All relevant data are within the paper and its Supporting Information files.

**Funding:** The author(s) received no specific funding for this work.

**Competing interests:** The authors have declared that no competing interests exist.

**Abbreviations:** SAAG, Serum-ascites albumin gradient; VA, Voltammetric analysis; PLS-DA, Partial Least Square Discriminant Analysis; PPV, Positive predictive value; NPV, Negative predictive value; WBC, White blood cell; PMN, Polymorphonuclear neutrophil count; SBP, Spontaneous bacterial peritonitis; LDH, Lactate dehydrogenase; E-TONGUE, Electronic tongue; WE, Working electrode; RE, Reference electrode; CE, Counter electrode; RMSECV, Root-mean-square-error cross validation; LR+, Positive likelihood ratio; LR-, Negative likelihood ratio.

portal hypertension and in 17/22 (77.2%) of cases with malignancy. Instead, VA showed poor predictive capacities towards total white blood count and polymorphonuclear cell count.

## Conclusions

According to this proof of concept study, VA qualifies as a promising low-cost and easy method to discriminate between ascites secondary to portal hypertension and ascites due to malignancy.

## Introduction

Ascites is the pathological accumulation of fluid in the peritoneal cavity. It is a consequence or complication of a number of diseases. Liver cirrhosis (75%) is the main cause of ascites in adults in the Western world, followed by malignancy (10%), heart failure (3%), tuberculosis (2%), and pancreatitis (1%).[1] Interestingly, approximately 5% of patients with ascites have two or more causes of ascites formation,[2] usually cirrhosis plus another cause, e.g., peritoneal carcinomatosis or peritoneal tuberculosis. The development of ascites in patients with cirrhosis is associated with a poor prognosis, as the five-year survival drops from about 80% in compensated cirrhotics to about 30% in patients with ascitic decompensation.[3] Generally, the presence of malignant ascites is a poor prognostic indicator, regardless of the cause,[4] with a median survival time ranging from 1 to 4 months.[5]

The diagnosis of newly-onset ascites is suspected on the basis of the history and physical examination and is usually confirmed by abdominal ultrasound. Abdominal paracentesis with fluid analysis is the first step in the evaluation of these patients. Laboratory tests in ascitic fluid initially include total and differential white blood cell (WBC) count, total protein, and albumin for calculation of the serum-ascites albumin gradient (SAAG).[6] To date, the serum-ascites albumin gradient (SAAG) is the most sensitive and specific marker for the differentiation of ascites due to portal hypertension from ascites due to other causes, with a diagnostic accuracy of 97%,[2] and it is recommended by the American and European guidelines.[6,7] SAAG is obtained by subtracting the level of albumin in the ascitic fluid from that in the serum. SAAG is low (<1.1 g/dL) in ascites not due to portal hypertension, such as in the case of malignancy, pancreatitis or infection. SAAG is high ($\geq$1,1 g/dL) in portal hypertension-related ascites, such as in liver cirrhosis or in congestive heart failure.[2,8] In patients with cirrhosis, whenever ascites is sampled, a total WBC count and differential should be obtained, as a polymorphonuclear neutrophil count (PMN) above 250 cells/µl is highly suspected for, and above 500 cells/µl is virtually diagnostic of spontaneous bacterial peritonitis (SBP).[9] According to clinical circumstances and pretest probability of specific disorders, other analytes, such as amylase, glucose, lactate dehydrogenase (LDH), triglycerides, etc., should be tested in the ascitic fluid. Finally, ascites can be cultured to detect the presence and type of infection, and, if malignancy is suspected, it can be sent to Pathology for cytological examination.

Research on possible new markers in the ascitic fluid, with the aim to simplify, improve or accelerate diagnostics, or to reduce the costs, has never stopped, and interesting findings have been reported in the last 20 years. In a study by Castellote J. *et al.*, a reagent strip for white blood cell (WBC) esterase designed for the testing of urine with a colorimetric 5-grade scale showed an optimal diagnostic accuracy for the diagnosis of SBP in cirrhotic patients.[10] However, subsequent results from a large multicenter series were not consistent with this finding.

[11] Ascitic fluid lactoferrin levels were reported as a useful diagnostic tool to identify SBP in cirrhotic patients with ascites, while elevated ascitic fluid lactoferrin in patients without SBP was suggested to be indicative of a developing hepatocellular carcinoma.[12] More recently, a panel of tumor markers determined in the ascitic fluid significantly increased the diagnostic performance of cytology for malignant ascites.[13]

Voltammetry is a powerful and versatile analytical technique that involves the application of a potential (*E*) to an electrode and the monitoring of the resulting current (*i*) flowing through the electrochemical cell. By careful interpretation of the voltammogram, important analytical information (quantitative and qualitative) is obtainable. Voltammetric analysis (VA) represents an exciting new development for the rapid analysis of biological fluids. This technique has been used for the detection of numerous electroactive species of biological significance such as urinary creatinine, urea, and alkaline ions,[14,15] as well as vitamins (thiamin, riboflavin, and pyridoxin),[16] hormons, and metals. Recently, VA has proved effective also for detection of urinary tract infections,[17,18] and leg ulcer infections.[19,20] Moreover, a similar technology for exhaled breath analysis (dubbed e-nose) has shown encouraging discriminatory capacities in different clinical scenarios.[21–24] However, VA has never been tested on the ascitic fluid, and the currently tested instrument is a prototype. Thus, in the absence of experience on the ascitic fluid, we founded our experiment on the encouraging diagnostic properties made evident on other biological fluids. Furthermore, the ascites has very heterogeneous composition depending upon its causes, and, thus, a method highly sensitive to changes in fluid composition like VA might have the potential for disclosing differences in etiology. Therefore, aim of this proof of concept work is to evaluate whether VA could add to the excellent discriminative capacity of SAAG vs portal hypertension by adding aetiologic information on ascites in a fast and inexpensive way.

## Material and methods

### Study design and population

Ascites samples were obtained from consecutive patients undergoing diagnostic or therapeutic paracentesis due to clinical indications at the Campus Bio-Medico Hospital of Rome, Italy, between November 2016 and May 2019. There were no exclusion criteria but, in the case of patients with repeated procedures, only the first ascites sample was processed. The study protocol was approved by the Ethics Committee of the University Campus Bio-Medico of Rome, and all participants signed an informed consent. There were no refusals to participate to the study.

### Specimens collection and analysis

Paracentesis was performed by means of a left iliac fossa puncture with a 22 gauge needle, and ascitic fluid was analyzed for its total WBC and differential, protein and albumin content, and LDH levels. In all cases, samples of ascites were cultured for aerobic and anaerobic bacteria and for Mycobacteria, and sent to the Pathology for cytological examination. On the same day of paracentesis, a blood drawn was performed in order to test for complete blood counts, LDH, serum protein and albumin.

Further investigations, e.g., abdominal ultrasound, CT scan or MR imaging, upper intestinal endoscopy, laparoscopy, laparotomy, etc., were carried out as deemed necessary in order to accomplish a definite diagnosis. Liver cirrhosis was diagnosed by a previous positive liver biopsy or clinically, by the combination of biochemical, ultrasonographic, elastometric and endoscopic findings. Congestive heart failure was diagnosed by clinical and echocardiographic findings. In patients with cirrhosis, SBP was identified by the presence of PMN $\geq 250/mm^3$ in

the ascitic fluid.[9] Malignant ascites was diagnosed by cytological findings in ascitic fluid, or by histological examination of samples obtained at surgery, usually but not necessarily in the presence of a clinical history of cancer.

An innovative multisensory system, the so-called electronic tongue (e-tongue), was applied on a sample of ascitic fluid. This sensor (BIONOTE) is intended to mimic the mechanism of human senses (BIOsensor-based multisensorial system for mimicking Nose, Tongue and Eyes). Basically, it consists of an information collecting unit for use in aqueous phase, connected to a routine for multivariate data processing. The liquid sensor array is made of three electrodes, respectively called working electrode (WE), reference electrode (RE), and counter electrode (CE). A potential excitation signal is applied between the WE and the RE; then, the current that flows between the working and the auxiliary electrodes is measured and the voltage signal is digitally acquired.[25] The input signal is a triangular function with a working range, from -1 to +1 V, thus resulting in 500 input voltages. The output signal is measured by the working electrode and is made up by the 500 corresponding output current values.[26] The electrode array consisting of silver, platinum and gold as working electrode (4 mm diameter), was fabricated by DropSens S.L. (Llanera, Spain). The experimental set-up is shown in Fig 1). Measurement time for each testing process required about 300 seconds. All output data were stored in a flash memory with separate names with date and time stampings for future references.

## Analytic approach

The characteristics of the study sample were reported as mean and standard deviation and median and interquartile range or absolute number and percentage for continuous and categorical variables, respectively. The ability of BIONOTE to discriminate ascitic fluid etiology and biochemical parameters was evaluated using Partial Least Square Discriminant Analysis (PLS-DA), with ten-fold cross-validation. Predictive capacities were expressed with the root-mean-square-error cross validation (RMSECV) to aggregate in a single measure of predictive power the magnitudes of the machine errors in prediction of continuous variables (total WBC count, PMN count, ascitic fluid albumin and SAAG). Conversely, for dichotomous outcomes (i.e., ascites secondary or not to portal hypertension), sensitivity, specificity, overall accuracy, positive (PPV) and negative (NPV) predictive values, positive (LR+) and negative (LR-) likelihood ratios were computed. All the analyses were performed using R version 3.3.0 (The R Foundation for Statistical Computing, Vienna, Austria).

## Results

General characteristics of study population are detailed in Table 1). Mean age was 68.6 years (SD: 12.5) and 46 patients (58%) were male. Overall, ascites was secondary to only portal hypertension in 72.5% of cases (58 subjects), while in the remaining cases (22 subjects) it was secondary to a neoplastic disease. Actually, in 6 of these latter cases, ascites was due to the combination of malignancy and portal hypertension (2 hepatocellular carcinoma and 2 cholangio-carcinoma on cirrhotic liver; 1 cervical cancer and 1 pancreatic cancer both complicated by portal thrombosis). In the analysis, these cases with mixed etiology were classified as neoplastic ascites for two main reasons: 1) primarily, because, from the prognostic point of view, the diagnosis of cancer stands above that of portal hypertension, and the risk of missing this information is clinically dangerous; 2) secondly, because, in the case of cancer, the mechanisms which leads to the leak of fluid are additional to that acting in pure portal hypertension. Ovarian cancer was the most prevalent cause of malignant ascites (9), followed by pancreatic cancer (3). Remaining etiologies of neoplastic effusions were mesothelioma (2), hepatocellular carcinoma

## Experimental Set-up

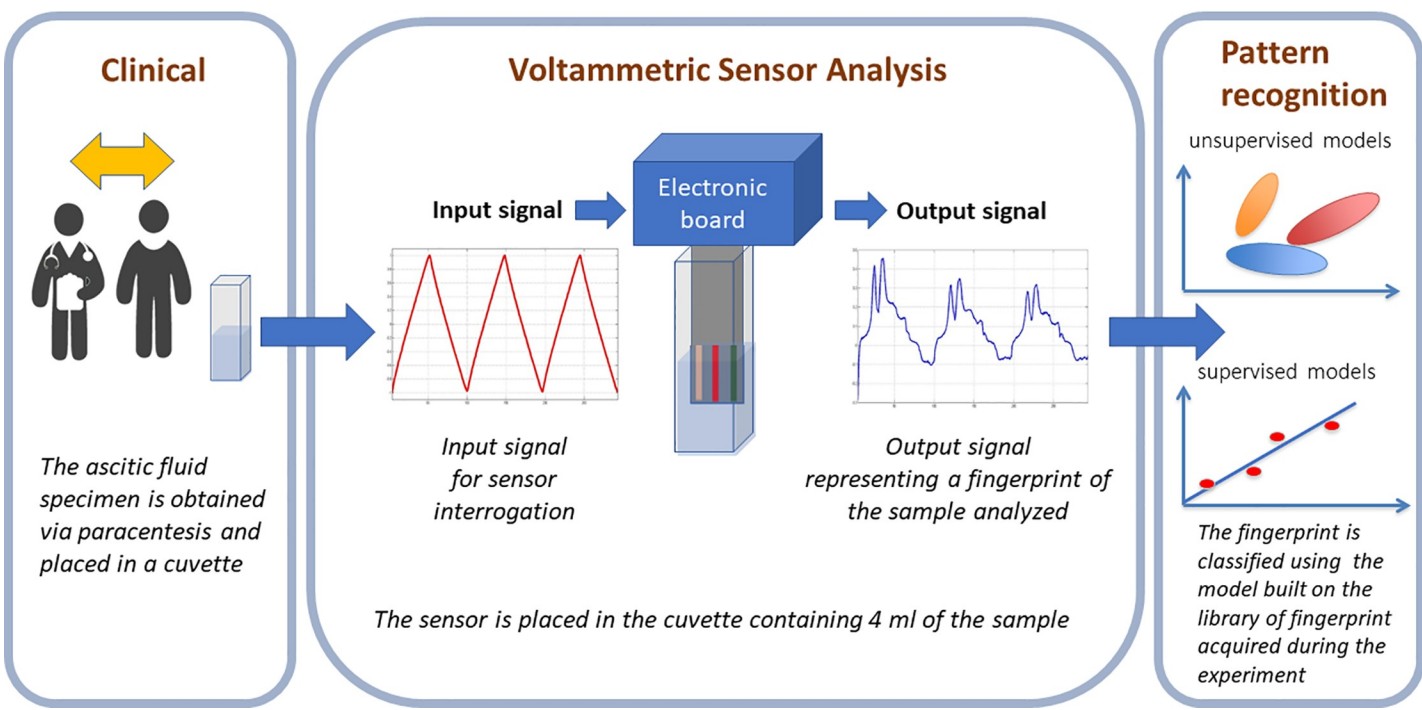

**Fig 1. Flow chart of the experimental set-up, illustrating the steps of the measurement process.** First step: sampling of the ascitic liquid from the patient. Second step: sample analysis via voltammetric sensor providing a characteristic fingerprint. Third step: fingerprint comparison with a model stored in a flash memory for the estimation of typical parameters of the sample.

(2), biliary duct adenocarcinoma (2), colonic carcinoma (2), endocervical adenocarcinoma (1) and cancer of unknown primary (1). Among patients with liver cirrhosis, 27 cases (46.5%) were secondary to viral hepatitis, 15 (25.8%) to non-alcoholic steatohepatitis, 11 (18.9%) to alcoholic liver disease and 5 (8.6%) to congestive hepatopathy. Among patients with liver cirrhosis, 4 received diagnosis of SBP. Main comorbidities were arterial hypertension (40%), diabetes mellitus (32.5%) and chronic kidney failure (23.7%).

Given the fully original application of VA, we had no preliminary information allowing to formally compute the sample size. However, based on the favorable performance of VA in different settings,[14–20] we assumed a number of 80 as functional to an exploratory analysis. Indeed, VA could satisfactorily disclose the infectious or non infectious state of cutaneous ulcers in only 25 patients and of the urine in 142 patients.[18,19]

E-tongue showed rather poor predictive capacities towards both total WBC count and PMN count both in the whole population (RMSECV of 485 and of 846 for total cell and PMN count, respectively; Fig 2, panel A) and limited to the subgroup of cirrhotic patients (RMSECV of 384 and of 236 for total cell and PMN count, respectively; Fig 2, panel B). Since only 4 patients were finally diagnosed with SBP, the diagnostic accuracy of e-tongue towards SBP was not calculated.

E-tongue predicted ascitic albumin with a RMSECV of 1.16 (Fig 3, panel A), and SAAG with a RMSECV of 0.99 (Fig 3, panel B). Fig 4) shows the average cyclic voltammograms of patients with ascites secondary to portal hypertension or neoplastic disease. When aiming to distinguish patients with ascites only due to portal hypertension from those with ascites due to a neoplastic disease, e-tongue correctly classified ascites etiology in 57/58 (98.2%) of cases with

**Table 1. General characteristics of study population.**

| Variable | Total sample |
|---|---|
| N | 80 |
| Age (years), mean (SD) | 68.6 (12.5) |
| Sex (male), n (%) | 46 (58) |
| Comorbidity | |
| *Hypertension, n (%)* | 32 (40) |
| *Diabetes Mellitus, n (%)* | 26 (32.5) |
| *Chronic Kidney Failure, n (%)* | 19 (23.7) |
| Etiology of ascites | |
| *Liver cirrhosis, n (%)* | 58 (72.5) |
| *Neoplastic, n (%)* | 22 (27.5) |
| Etiology of liver disease | |
| *Viral, n (%)* | 27 (46.5) |
| *Alcoholic, n (%)* | 11 (18.9) |
| *Non-alcoholic, n (%)* | 15 (25.8) |
| *Cardiogenic, n (%)* | 5 (8.6) |
| Child-Pugh Class | |
| *A, n (%)* | 4 (6.8) |
| *B, n (%)* | 37 (63.7) |
| *C, n (%)* | 17 (29.3) |
| Serum Albumin (g/dL), mean (SD) | 2.8 (0.6) |
| Ascites Albumin (g/dL), mean (SD) | 1.1 (0.8) |
| SAAG, mean (SD) | 1.7 (0.9) |
| Total cell count, median (IQR) | 297 (135–603) |
| PMN count, median (IQR) | 26 (8–90) |
| Positive ascites cultures, n (%) | 1 |

SD, standard deviation; n, number; IQR, interquartile range

portal hypertension and in 17/22 (77.2%) of cases with malignant ascites. Compared to SAAG≥1.1 g/dL, e-tongue predicted ascites only due to portal hypertension with a better accuracy (92.5% *Vs* 87.5%); sensitivity (98.3% *Vs* 94.8%); specificity (77.3% *Vs* 68.2%); predictive values (PPV 91.9% *Vs* 88.7% and NPV 94.4% *Vs* 83.3%). Data are summarized in Table 2).

Overall, SAAG and e-tongue results were discordant in 14 cases, 7 because SAAG was <1.1 g/dl and e-tongue was diagnostic for portal hypertension, and 7 because SAAG was ≥1.1 g/dl and e-tongue was diagnostic for malignancy. In the seven discordant cases in which SAAG was <1.1 g/dl, SAAG correctly classified ascites as due to malignancy in 4 cases while e-tongue was right in the other 3; in the seven discordant cases in which SAAG was ≥1.1 g/dl, SAAG missed the diagnosis of malignancy in 5 cases, all of which were correctly classified by e-tongue, while SAAG correctly classified 2 cases of pure portal hypertension in cirrhosis. Notably, in the 6 cases of patients with ascites due to malignancy but with concurrent portal hypertension, SAAG was always diagnostic for the condition of portal hypertension (≥1.1 g/dl), while e-tongue recovered the diagnosis of malignancy in 4 of the cases.

## Discussion

In the present study, the evaluation of electrochemical patterns in solution using BIONOTE seems to outperform SAAG in distinguishing ascites due to portal hypertension from ascites

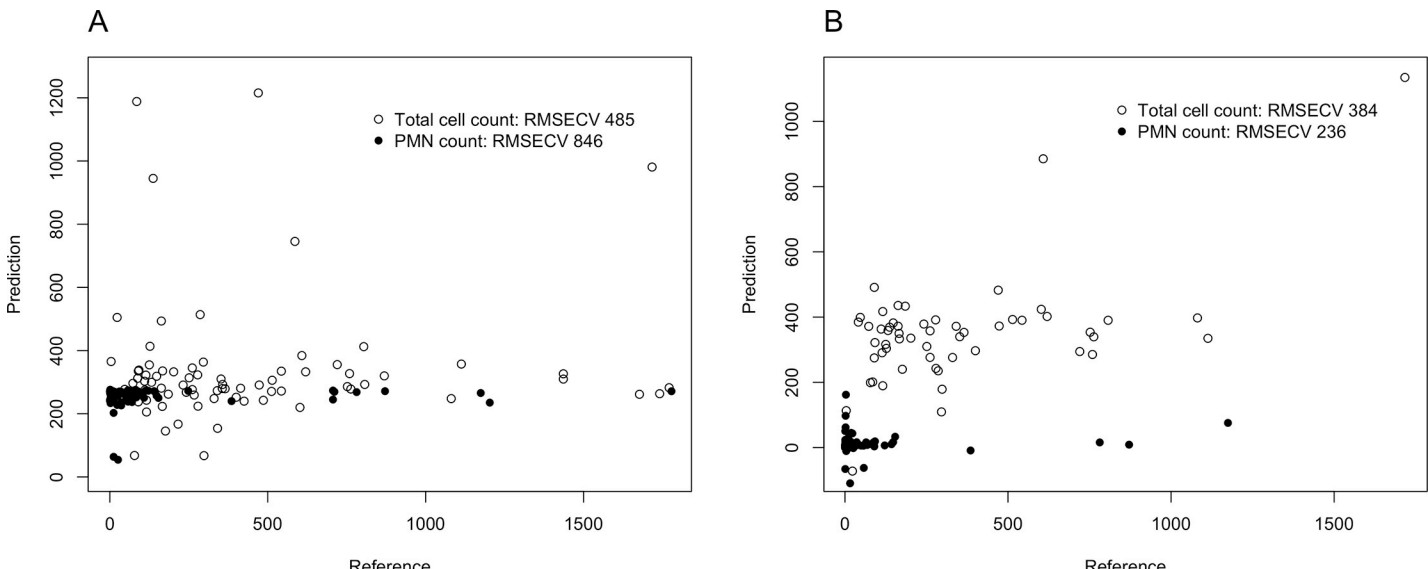

**Fig 2.** Prediction of total WBC and PMN count in the whole population (panel A) and in the group of cirrhotic patients (panel B), according to VA by e-tongue.

due to malignancy. Conversely, the diagnostic accuracy of voltammetry with respect to ascites PMN count was unsatisfactory, and VA seems less promising for the screening of SBP.

The enormous diagnostic potential of analyzing the ascitic fluid has long been known. Excluding the less frequent determination of analytes which can be useful in specific conditions, such as triglycerides in chylous ascites or amylases in pancreatic ascites, the ascitic fluid is frequently tested for its albumin content and PMN count, in order to discriminate between portal hypertension and other causes of ascites, and to verify whether the ascitic fluid is infected or not in patients with cirrhosis, respectively.

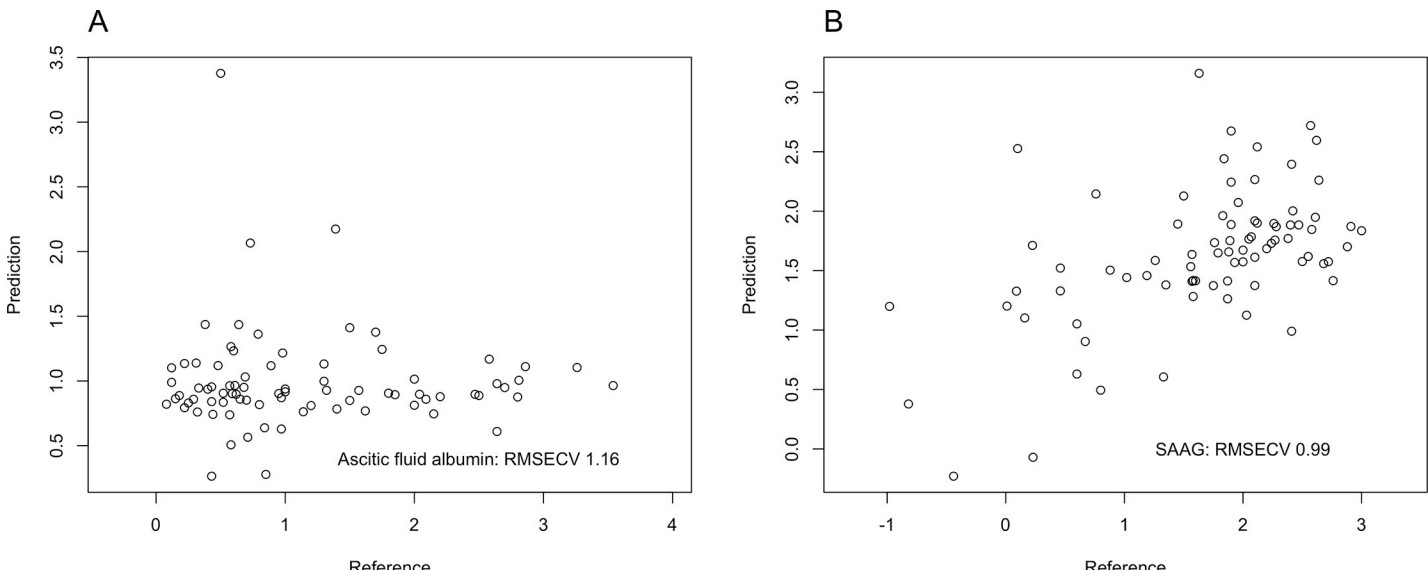

**Fig 3.** Prediction of ascitic albumin (panel A) and SAAG (panel B), according to VA by e-tongue.

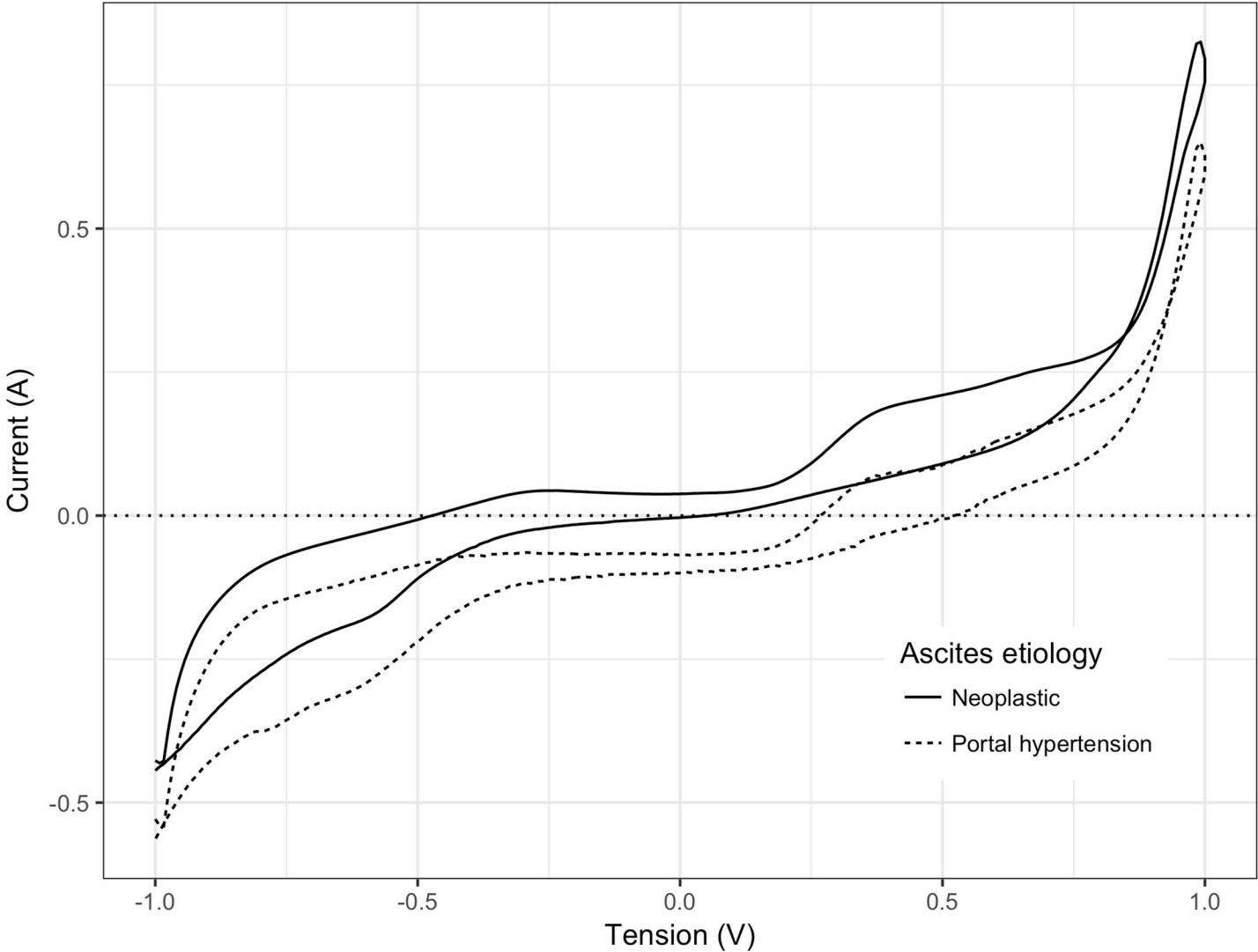

**Fig 4.** Cyclic voltammogram of ascites secondary to portal hypertension (dashed line) or to neoplastic disease (solid line).

**Table 2. Diagnostic performances of SAAG> = 1.1 or VA by e-Tongue for diagnosis of ascites secondary to portal hypertension or to neoplastic disease.**

|  | SAAG> = 1.1 | VA |
|---|---|---|
| **Sensitivity** | 94.6 (85.1–98.9) | 98.2 (90.4–100) |
| **Specificity** | 62.5 (40.6–81.2) | 70.8 (48.9–87.4) |
| **Accuracy** | 85 (75.3–92) | 90 (81.2–95.6) |
| **PPV** | 85.5 (74.2–93.1) | 88.7 (78.1–95.3) |
| **NPV** | 83.3 (58.6–96.4) | 94.4 (72.7–99.9) |
| **LR+** | 2.5 (1.5–4.2) | 3.4 (1.8–6.3) |
| **LR-** | 0.09 (0.03–0.27) | 0.03 (0.004–0.2) |

VA, voltammetric analysis; SAAG, serum-ascites albumin gradient; PPV, positive predictive value; NPV, negative predictive value; LR, likelihood ratio

Cirrhosis is by far the most common cause of ascites, and the main differential diagnosis is that with malignant ascites, which represents 10% of the cases of ascites.[1,2] Notably, Runyon et al. found that the accuracy of SAAG to distinguish between ascites due to portal hypertension and other causes approached 97%.[2] However, it has been demonstrated that SAAG may be falsely low in patients with ascites due to portal hypertension in the presence of hypoalbuminemia, systemic hypotension, and hypergammaglobulinemia (> 5 g/dl).[27–29] A falsely high value of SAAG may occur in chylous ascites as lipid fractions tend to interfere with laboratory determination of albumin.[2] Actually, the most insidious condition in which SAAG may be misleading is that in which portal hypertension and cancer are concurrently present. This is, for example, the common case of hepatocellular or cholangiocarcinoma on the background of cirrhotic liver. However, this is not infrequent also in other types of cancer. Indeed, in the work by Chen *et al.*, up to 38% of patients with malignant ascites could have a SAAG $\geq$ 1.1 g/dL because of portal hypertension as a result of massive hepatic metastasis or portal vein thrombosis.[30] In all these cases with mixed etiology, SAAG "feels" only portal hypertension and misses the much more important diagnosis of concurrent malignancy. Notably, here we observed that, in the 6 cases of patients with ascites due to malignancy but with concurrent portal hypertension, SAAG was always diagnostic for the condition of portal hypertension ($\geq$1.1 g/dl), while e-tongue recovered the diagnosis of malignancy in 4 of the cases. According to the Starling hypothesis, the fluid movement across a capillary membrane is controlled by the balance of hydrostatic and colloid osmotic forces across the capillary wall. [31] Since albumin is the main determinant of oncotic pressure, SAAG correctly reflects the presence or absence of portal hypertension in the genesis of ascites. However, malignant tumors can cause effusions also by different mechanisms which include increasing the permeability of the serosal surface or obstructing draining lymphatics.[32,33] Thus, it might allow the passage of plasma substances different from albumin, with electrochemical properties, which are missed by SAAG but could be detected by VA.

Conversely, e-tongue did not show a good sensitivity for ascitic fluid WBC and PMN count. Actually, recent studies have reported a good diagnostic performance of VA for detecting infections of biological fluids. Lelli *et al.* tested VA in urine and compared results with those obtained by dipstick. Overall, VA showed an accuracy of 81.7% (95% CI 74.3–87.7%) in detecting urinary tract infections with respect to 75.9% (95% CI 68–82.7%) displayed by dipstick.[18] Similarly, VA showed an overall accuracy of 94% in detecting infections of leg ulcers. [19] Currently, we do not have a clear explanation for the reduced diagnostic performance of e-tongue with respect to WBC count in ascites.

To our knowledge, this is the first study reporting on VA in ascitic fluid. The main advantage of this modern voltammetric method is that, although the specificity of each sensor is low, the combination of several specificity classes entails a very large information potential and the combination of all electrodes generates a unique fingerprint.[34] Thus, the analysis of electrochemical properties of a given fluid may account upon a huge variety of experimental conditions, i.e., broadly speaking, of experimental electrical fields. By identifying the experimental conditions yielding the best discriminatory capacity it will be possible to further simplify the experimental procedure as well as to increase its diagnostic accuracy. Furthermore, a more detailed characterization of ascites will allow verify whether a given e-tongue finding, i.e., a well-defined electrochemical pattern, recognizes selected determinants and, thus, points at a distinctive chemical composition. [25,35] Finally, portal hypertension is a heterogeneous condition as for etiology and severity, and neoplastic ascites has many determinants and variable physicochemical properties. Thus, expanding the library to compare different and well-sized categories of either hemodynamic or neoplastic ascites is expected to provide potential insight into the diagnostic properties of the e-tongue. In addition to this, VA is a quick technique

(about 10 minutes), with relatively low-cost, about 7 dollars for exam, due to the minimal instrumental requirements and ease of application.

The present study has an important limitation. Indeed, it was carried out in a relatively small sample of patients, with a prevalence of ascites due to portal hypertension, and the group of cirrhotics with SBP was limited. Furthermore, it needs to be validated in a testing population. However, this study has also some strengths. Firstly, it is highly original, being the first study to explore the diagnostic potential of VA in ascites. Secondly, it included only non-repetitive cases, optimally characterized by the diagnostic point of view, thus giving a clear answer to the proof-of-concept question it was designed for. Finally, the operators caring for a given diagnostic assays (SAAG, VA, others) were fully blinded to patient's characteristics and results of the remaining assays.

## Conclusion

SAAG remains the most sensitive and specific marker for the differentiation of ascites due to portal hypertension from ascites due to other causes. However, SAAG has some limitations and may fail in selected conditions; among these, missing the diagnosis of neoplastic effusion is of particular clinical relevance. VA of ascitic fluid seems a very promising method to discriminate between ascites due to portal hypertension and ascites due to malignancy. Further investigation is clearly awaited in order to confirm and extend these findings towards a concrete clinical application. Furthermore, the follow-up of these patients will allow disclose prognostic properties of the e tongue, e.g. inherent to the responsivity to diuretics or major outcomes. If these preliminary results will be confirmed, VA could integrate and complement the SAAG in the diagnostic work up of ascites.

## Supporting information

**S1 Data.**
(XLSX)

## Author Contributions

**Conceptualization:** Moises Muley, Umberto Vespasiani-Gentilucci, Marco Santonico, Giorgio Pennazza, Antonio Picardi, Raffaele Antonelli-Incalzi.

**Data curation:** Moises Muley, Antonio De Vincentis, Marco Santonico, Giorgio Pennazza, Simona Sanguedolce, Cristiana De Luca, Francesco Plotti.

**Formal analysis:** Moises Muley, Antonio De Vincentis, Marco Santonico, Giorgio Pennazza, Cristiana De Luca.

**Investigation:** Moises Muley, Umberto Vespasiani-Gentilucci, Simona Sanguedolce.

**Methodology:** Moises Muley, Antonio De Vincentis, Marco Santonico, Giorgio Pennazza, Simona Sanguedolce, Francesco Plotti, Antonio Picardi, Raffaele Antonelli-Incalzi.

**Project administration:** Umberto Vespasiani-Gentilucci, Raffaele Antonelli-Incalzi.

**Resources:** Raffaele Antonelli-Incalzi.

**Software:** Moises Muley, Antonio De Vincentis, Marco Santonico.

**Supervision:** Umberto Vespasiani-Gentilucci, Raffaele Antonelli-Incalzi.

**Validation:** Umberto Vespasiani-Gentilucci, Antonio De Vincentis, Raffaele Antonelli-Incalzi.

**Visualization:** Moises Muley.

**Writing – original draft:** Moises Muley, Umberto Vespasiani-Gentilucci.

**Writing – review & editing:** Moises Muley, Umberto Vespasiani-Gentilucci, Raffaele Antonelli-Incalzi.

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
