## [Decision Letter · Decision Letter 0]

14 Apr 2020

PONE-D-20-09873

Voltammetric analysis for distinguishing portal hypertension-related from malignancy-related ascites: a proof of concept study.

PLOS ONE

Dear M.D. De Vincentis,

Thank you for submitting your manuscript to PLOS ONE. After careful consideration, we feel that it has merit but does not fully meet PLOS ONE’s publication criteria as it currently stands. Therefore, we invite you to submit a revised version of the manuscript that addresses the points raised by the reviewers. In addition to that, it is important that you make sure that your report meets the criteria of utility validation and availability described in: http://journals.plos.org/plosone/s/submission-guidelines#loc-methods-software-databases-and-tools, and the STROBE gudelines/checklist: http://www.strobe-statement.org.

We would appreciate receiving your revised manuscript by May 29 2020 11:59PM. To enhance the reproducibility of your results, we recommend that if applicable you deposit your laboratory protocols in protocols.io, where a protocol can be assigned its own identifier (DOI) such that it can be cited independently in the future. For instructions see: http://journals.plos.org/plosone/s/submission-guidelines#loc-laboratory-protocols

We look forward to receiving your revised manuscript.

Kind regards,

Matias A Avila, Ph.D.

Academic Editor

PLOS ONE

Reviewers' comments:

Reviewer's Responses to Questions

**Comments to the Author**

1. Is the manuscript technically sound, and do the data support the conclusions?

Reviewer #1: Yes

Reviewer #2: Yes

2. Has the statistical analysis been performed appropriately and rigorously? 

Reviewer #1: Yes

Reviewer #2: Yes

3. Have the authors made all data underlying the findings in their manuscript fully available?

Reviewer #1: Yes

Reviewer #2: Yes

4. Is the manuscript presented in an intelligible fashion and written in standard English?

Reviewer #1: Yes

Reviewer #2: Yes

5. Review Comments to the Author

Reviewer #1: I read with interest this manuscript. The study by Antonio De Vincentis described Voltammetric analysis for distinguishing portal hypertension-related from malignancy- related ascites. The manuscript is well written and presented, conclusions are sound. The topic is of interest to the scientific community.

I have only one minor revision:

-The authors should discussion in the conclusion how they intend to further improve the accuracy of this technique in future studies. This is a proof of concept study and I think that is important to give future prospects for studies.

Reviewer #2: The paper its related to a study, through Voltammetric analysis, for distinguishing portal hypertension-related from malignancy related ascites. The paper is interesting, but authors should convince the reviewer that the proposed method is valid, by comparing it with the literature. Moreover, better information on experimental set-up (BioNote), in particular, technical data and some figures related to measurements, must be given for a broader audit of the paper (sensor researchers).

6. PLOS authors have the option to publish the peer review history of their article (what does this mean?). If published, this will include your full peer review and any attached files.

Reviewer #1: Yes: Andrea Casadei Gardini

Reviewer #2: No

---

## [Author Response · Author response to Decision Letter 0]

2 May 2020

Answer to the comments

Reviewer #1: I read with interest this manuscript. The study by Antonio De Vincentis described Voltammetric analysis for distinguishing portal hypertension-related from malignancy-related ascites. The manuscript is well written and presented, conclusions are sound. The topic is of interest to the scientific community. I have only one minor revision: -The authors should discussion in the conclusion how they intend to further improve the accuracy of this technique in future studies. This is a proof of concept study and I think that is important to give future prospects for studies.

Answer to the comment: We are glad the reviewer appreciated our work and we thank him/her for the comment. The analysis of electrochemical properties of a given fluid may account upon a huge variety of experimental conditions, i.e., broadly speaking, of experimental electrical fields. [26] Thus, comparing different experimental conditions to identify that or those yielding the best discriminatory capacity is expected to further simplify the experimental procedure as well as to increase its diagnostic accuracy. Furthermore, a more detailed characterization of ascites will allow verify whether a given e-tongue finding, i.e., a well-defined electrochemical pattern, recognizes selected determinants and, thus, points at a distinctive chemical composition. Finally, portal hypertension is a heterogeneous condition as for etiology and severity, and neoplastic ascites also has many determinants and variable physicochemical properties. Thus, expanding the library to compare different and well-sized categories of either hemodynamic or neoplastic ascites is expected to provide potential insight into the diagnostic properties of the e-tongue. Indeed, we cannot exclude that selected clinical and pathophysiological conditions will benefit the most from the classificatory and prognostic properties of the e-tongue. In this perspective, the follow up of patients might be of interest to test the prognostic properties of the voltammetric findings versus outcomes such as responsivity of ascites to diuretic therapy or survival. These considerations are now available in the “Discussion” section on pages 13 to 14, lines 290 to 300, and in the “Conclusion” section on page 14, lines 319 to 321.

Reviewer #2: The paper its related to a study, through Voltammetric analysis, for distinguishing portal hypertension-related from malignancy related ascites. The paper is interesting, but authors should convince the reviewer that the proposed method is valid, by comparing it with the literature. Moreover, better information on experimental set-up (BioNote), in particular, technical data and some figures related to measurements, must be given for a broader audit of the paper (sensor researchers).

Answer to the comment: We thank the reviewer for his/her comment. Concerning the comparison of BIONOTE with scientific literature on the field, it is worth remarking that no experiment using this sensor in ascitic fluid has been reported so far. However, we had previously tested BIONOTE for classification of other biological fluid. [18-20] In the revised manuscript, the measurement process using BIONOTE has been illustrated in a figure, i.e., Figure 1) to make it clearer to any kind of reader. Generally, chemical sensors are based on specific or non-specific sensing materials. The kind of sensing material could contribute to obtain selective or non-selective sensors. Accordingly, Biosensors are specific sensors and their sensing mechanisms is so called “key-lock”. Using a sensing material on the transducer requires particular processes of manufacturing in order to obtain reproducible and repeatable sensors. In our case we used a non-functionalized sensor, in particular no sensing materials are used on the electrode surface. 

The sensing material has an advantage for bio-medical applications (in general), but two disadvantages for the specific application here presented. The advantage consists in the possibility of selecting a well defined sensing element interacting with a specific biomarker or a specific compound of interest for the sample under test, increasing sensor performance in terms of sensitivity, resolution and, obviously, selectivity. However, the approach of this paper is the fingerprinting and not the identification and/or quantification of specific biomarkers. Moreover, the process of covering the surface of the sensor with a chemical interactive material requests another technological step which affects the overall reproducibility of the sensor.

The commercial electrode (Au, Pt, Ag) with a dedicated electronic interface developed by the Unit of Electronics for Sensor Systems of Campus Bio-Medico di Roma guarantees the reproducibility and repeatability of measurement as shown in other papers [25,26,35]. In the future this kind of approach could reduce the operational costs of the device. We did not include these comments in the text because they seems more suited for a pure technological journal. However, the bases of the voltammetric analyses are reported in detail in the above cited references and in the following three.

---

## [Editor Report · Decision Letter 1]

5 May 2020

Voltammetric analysis for distinguishing portal hypertension-related from malignancy-related ascites: a proof of concept study

PONE-D-20-09873R1

Dear Dr. De Vincentis,

We are pleased to inform you that your manuscript has been judged scientifically suitable for publication and will be formally accepted for publication once it complies with all outstanding technical requirements.

With kind regards,

Matias A Avila, Ph.D.

Academic Editor

PLOS ONE
---

## [Editor Report · Acceptance letter]

13 May 2020

PONE-D-20-09873R1 

Voltammetric analysis for distinguishing portal hypertension-related from malignancy-related ascites: a proof of concept study 

Dear Dr. De Vincentis:

I am pleased to inform you that your manuscript has been deemed suitable for publication in PLOS ONE. Congratulations! Your manuscript is now with our production department. 

With kind regards,

on behalf of

Dr Matias A Avila 

Academic Editor

PLOS ONE